# Costs and cost-effectiveness of treatment setting for children with wasting, oedema and growth failure/faltering: A systematic review

Noreen Dadirai Mdege[1,2]*, Sithabiso D. Masuku[3], Nozipho Musakwa[3], Mphatso Chisala[4], Ernest Ngeh Tingum[5], Micheal Kofi Boachie[6], Farhad Shokraneh[7]

1 Department of Health Sciences, University of York, York, United Kingdom, 2 Centre for Research in Health and Development, York, United Kingdom, 3 Health Economics and Epidemiology Research Office, Faculty of Health Sciences, University of the Witwatersrand, Johannesburg, South Africa, 4 Department of Population, Policy and Practice, Great Ormond Street Hospital, Institute of Child Health, University College London, London, United Kingdom, 5 Department of Economics, University of Namibia, Windhoek, Namibia, 6 Discipline of Public Health Medicine, School of Nursing and Public Health, University of KwaZulu-Natal, Durban, South Africa, 7 Department of Evidence Synthesis, Systematic Review Consultants LTD, Nottingham, United Kingdom

* noreen.mdege@york.ac.uk

**Data Availability Statement:** The data for this systematic review can be accessed on: Mdege, Noreen Dadirai, Masuku, Sithabiso, Musakwa,

## Abstract

This systematic review aimed to address the existing evidence gaps, and guide policy decisions on the settings within which to treat infants <12 months of age with growth faltering/failure, and infants and children aged <60 months with moderate wasting or severe wasting and/or bilateral pitting oedema. Twelve electronic databases were searched for studies published before 10 December 2021. The searches yielded 16,709 records from which 31 studies were eligible and included in the review. Three studies were judged as low quality, whilst 14 were moderate and the remaining 14 were high quality. We identified very few cost and cost-effectiveness analyses for most of the models of care with the certainty of evidence being judged at very low or low. However, there were 17 cost and 6 cost-effectiveness analyses for the initiation of treatment in outpatient settings for severe wasting and/or bilateral pitting oedema in infants and children <60 months of age. From this evidence, the costs appear lowest for initiating treatment in community settings, followed by initiating treatment in community and transferring to outpatient settings, initiating treatment in outpatients then transferring to community settings, initiating treatment in outpatient settings, and lastly initiating treatment in inpatient settings. In addition, the evidence suggested that initiation of treatment in outpatient settings is highly cost-effective when compared to doing nothing or no programme implementation scenarios, using country-specific WHO GDP per capita thresholds. The incremental cost-effectiveness ratios ranged from $20 to $145 per DALY averted from a provider perspective, and $68 to $161 per DALY averted from a societal perspective. However, the certainty of the evidence was judged as moderate because of comparisons to do nothing/ no programme scenarios which potentially limits the applicability of the evidence in real-world settings. There is therefore a need for evidence that compare the different available alternatives.

Nozipho, & Chisala, Mphatso. (2023). Dataset for a systematic review on the costs and cost-effectiveness of treatment setting for children with wasting, oedema and growth failure/faltering [Data set]. Zenodo. https://doi.org/10.5281/zenodo.8002937.

**Funding:** This work was supported by the World Health Organization (Purchase order 202760343 to NDM, SDM, NM, MC, and FS). The funder had no role in study design, data collection and analysis, decision to publish, or preparation of the manuscript. The findings and conclusions in this report are those of the authors and do not necessarily represent the official position of the World Health Organization.

**Competing interests:** The authors have declared that no competing interests exist.

# Background

Child wasting, i.e., a child who is too thin for their height, can develop rapidly in the face of poor nutrient intake and/or disease [1, 2]. In 2020, about 6.7% (i.e., 45.4 million) of the world's children under 5 years of age were affected by wasting, and 13.6 million were severely wasted [1]. The prevalence of wasting is highest in low- and middle-income countries (LMICs), with the majority of cases being in Sub-Saharan Africa and South Asia [3]. Moderately or severely wasted children have a weakened immunity, are susceptible to long-term developmental delays, and have a 5- to 20-fold increased risk of death [2–4]. Globally, each year, about 4.4% of deaths among children under 5 years of age are attributable to severe wasting [5]. Growth faltering/failure, on the other hand, describes lower weight or rate of weight gain, lower height, or an abnormally slow rate of gain in a child's height or length, than expected for age and sex in childhood [6–8]. In LMICs the rates of early-life growth faltering/failure are unacceptably high due to poor health and social conditions [6, 7].

The current World Health Organization (WHO) guidelines for severe acute malnutrition (also referred to as severe wasting and oedema) in infants and children [2] have several gaps, including on recommendations for growth failure/faltering in infants under 6 months of age, the management of moderate wasting, and economic evidence to support decision making. Reviews of interventions for growth faltering/failure and child wasting in infants and young children have largely focused on the health and human impacts [9]. The few cost-effectiveness reviews that currently exist have focused on the different child undernutrition treatments, and the treatment of moderate or severe acute malnutrition at the community level [9, 10]. There is a glaring gap in reviews of cost-effectiveness evidence to guide policy decisions on the settings within which to treat this population.

We conducted this WHO-commissioned systematic review in order to complement the existing evidence and strengthen the WHO guidelines and recommendations. The review evaluates costs and cost-effectiveness of: initiation of treatment in a community setting; initiation of treatment in outpatient settings; referral to treatment from community to outpatient settings; referral to treatment in an inpatient setting; transfer from inpatient to outpatient/community treatment; transfer from outpatient to community settings; and discharge from outpatient/community treatment. The review focused on: 1) infants <12 months of age with growth faltering/failure; and 2) infants and children aged <60 months with moderate wasting or severe wasting and/or bilateral pitting oedema.

# Materials and methods

The systematic review was guided by well-established standardised principles and methods, including a pre-written protocol [11, 12]. The protocol was not registered, but was peer-reviewed by child nutrition, health economics and systematic review experts, and published [13]. The PRISMA Checklist is provided as S1 File.

## Inclusion/ Exclusion criteria

The inclusion and exclusion criteria are detailed in Table 1. Definitions used in this review for the population (i.e., moderate wasting, severe wasting and/or bilateral pitting oedema and growth failure/faltering), settings (i.e., community, outpatient and inpatient settings), and type of care (i.e., treatment initiation, referral, transfer or discharge) are provided in S2 File.

**Table 1. Inclusion/ Exclusion criteria.**

| Selection Criteria | Inclusion | Exclusion |
|---|---|---|
| Population | Infants and children <5 years of age with moderate wasting or severe wasting and/or bilateral pitting oedema.<br>Infants <12 months of age with growth failure/ faltering | For moderate and severe wasting and/or oedema<br>Mixed populations that include the population of interest (i.e., infants and children <5 years of age with moderate or severe wasting and/or oedema) but where data for the population of interest is not reported separately.<br>For growth failure/ faltering:<br>Mixed populations that include the population of interest (i.e., infants <12 months of age with growth failure/ faltering), but where data for the population of interest is not reported separately. |
| Intervention | For wasting or growth failure/faltering:<br>• initiation of treatment in a community setting.<br>• initiation of treatment in outpatient settings.<br>• referral to treatment in an inpatient setting.<br>• transfer from inpatient to outpatient/ community treatment<br>• discharge from outpatient/community treatment<br>. | Other interventions that are not those listed in the inclusion criteria. |
| Comparators | Not restricted (with or without a comparator) | N/A |
| Outcomes | Resource use<br>Costs<br>Cost-effectiveness estimates based on a) cost outcome analysis (e.g., cost per child seen etc.), or b) full cost-effectiveness analysis (e.g., cost per life years saved etc.). | • Only indirect costs reported, such as productivity loss.<br>• Only including costs of medicinal food with no setting-related costs |
| Study type | Any type of economic analysis (including cost and cost-effectiveness or cost-utility analyses) reporting cost estimates based on a) patient-level data, b) expenditure or c) ingredients, or a combination thereof, or calculating costs based on treatment pathways in clinical guidelines | Systematic reviews and other types of literature reviews to avoid double counting |
| Language | No restrictions | N/A |
| Other | Studies that are available as full text | Publications which do not report relevant outcomes (e.g., study protocols, commentaries and letters for the Editor) |

## Search strategy

The following databases were searched on 10th December 2021: Global Health Cost Effectiveness Analysis Registry and the Cost-Effectiveness Analysis Registry via the Center for the Evaluation of Value and Risk in Health; Cochrane Central Register of Controlled Trials (CENTRAL) and Cochrane Database of Systematic Reviews via Cochrane Library; CRD's NHS Economic Evaluation Database and HTS Database (available only until 2015); EconLit via ProQuest Dialog; Embase via Ovid SP; Epistemonikos; Google Scholar (including Grey Literature); INAHTA HTA Database; and Ovid MEDLINE ALL. The search terms were selected from experts' opinions, literature review, reviewing the results of scoping searches, and controlled vocabularies (Medical Subject Heading = MeSH and Excerpta Medica Tree = Emtree). The terms were arranged into three blocks: Block 1, terms for children/ infants; Block 2, terms for wasting or growth failure; and Block 3, terms for study design or outcomes (e.g., cost

analysis, cost-effectiveness etc). No date, study design, publication type, geographic or language limits were imposed on the searches. All search strategies are reported in S3 File.

We also searched the websites of Action Against Hunger, MSF, Save the Children, UNICEF, WHO, and the World Bank. Citations and reference lists of included publications and previous systematic reviews were also manually reviewed to identify additional literature.

## Study selection

The Rayyan software was used to manage the articles retrieved from the searches [14]. Each article was independently screened for eligibility by two independent reviewers using a piloted study screening form based on the inclusion/exclusion criteria. Titles and abstracts were screened during the first stage of study selection. Studies judged to be potentially eligible in the first stage had their full texts screened in the second stage.

## Data extraction

For each of the included studies, two reviewers independently extracted the relevant data using a standardised Microsoft Excel data extraction table. The table was piloted on five studies before use and adjusted accordingly [11, 12]. The extracted data included general information such as author, publication year, country, WHO region; study methodology; population; details of intervention; and outcomes (for more details see S4 File).

## Quality assessment strategy

The included studies were published between 1972 and 2021. Methodological or reporting quality was assessed using the 2013 ISPOR Consolidated Health Economic Evaluation Reporting Standards (CHEERS), which is the guidance that was applicable during that period [15]. Each item on the checklist was graded for each study as follows: 0 (not considered), 1 (partially considered), 2 (fully considered) and N/A (if not relevant to the study). The item scores were subsequently summed up and a percentage calculated based on the maximum attainable score. Studies with a percentage score less than 50% were categorised as low, those with a score between 50% and 74% as moderate, and those with a score of 75% or higher as good quality studies. The Grading of Recommendations, Assessment, Development and Evaluations (GRADE) system and the UK National Institute for Health and Care Excellence's economic profiles approach were used to classify the certainty in the evidence across all studies as very low, low, moderate, or high [16–18]. First, we built an economic profile for the available evidence for each topic using the following criteria: resource allocation, cost-effectiveness evidence, overall quality of evidence, applicability, certainty and any other limitations (Table 2).

Evidence based on cost-effectiveness analysis was considered as high quality. For each model of care, each of the criteria above was given a rating [16]. If no serious concern existed for any of these criteria, the recommendation was not downgraded; if serious concern existed for at least one of the criteria, the evidence was downgraded one level (-1), e.g., from high to moderate. In the case of very serious concern for at least one of the criteria, the downgrade was two levels (-2), e.g., from high to low. Evidence that was only based on cost analysis was considered as low quality, with upgrades (i.e., +1 or +2) for large effect, dose-response, or no confounding.

We used the GRADE definitions where the quality of the evidence was considered as [16].

- high when there was strong confidence that the true value lies close to the estimated value,

- moderate when the true value was likely to be close to the estimated value, but there was a possibility that it was substantially different,

**Table 2. Criteria for economic profiles.**

| Criteria | Considerations | Rationale for judgement |
|---|---|---|
| Resource allocation | • number of studies reporting the costs of an intervention<br>• how the costs compare with other models of care | • the higher the costs of one model of care compared to the alternatives, the lower the likelihood that a strong recommendation was warranted [16].<br>• The higher the number of studies reporting consistent results, the higher the likelihood of a strong recommendation. |
| Cost-effectiveness evidence | • number of studies reporting the costs of an intervention and the incremental cost-effectiveness ratio when compared with other models of care against the appropriate threshold | • if an intervention is cost-effective compared to the alternatives, a strong recommendation is warranted.<br>• The higher the number of studies reporting consistent results, the higher the likelihood of a strong recommendation. |
| Overall quality of evidence | Based on the CHEERS checklist | • The higher the quality of the evidence, the higher the likelihood that a strong recommendation is warranted [16]. |
| Applicability | How well does the included evidence answer the review question [17]?<br>1. Are the study populations and the interventions being evaluated the same as those depicted in the review question?<br>2. Are the comparisons being made between real-life/ viable alternatives [18]? | • Directly applicable if the studies meet all applicability criteria or fail to meet one or more applicability criteria, but this is unlikely to change the conclusions about cost-effectiveness<br>• partially applicable if the studies fail to meet one or more of the applicability criteria, and this could change the conclusions about cost-effectiveness<br>• not applicable if the studies fail to meet one or more of the applicability criteria, and this is likely to change the conclusions about cost-effectiveness. |
| Certainty | The extent to which there was confidence that an estimate of an effect from the whole body of evidence was adequate to make a decision or a recommendation [18]? | The higher the confidence in the estimate, the higher the likelihood that a strong recommendation is warranted |
| Other limitations | Other limitations either identified in the study report itself, or by the reviewers. | What are the implications on the confidence in the estimates? |

- low when the true value could be substantially different from the estimated value,

- very low when the true value was likely to be substantially different from the estimated value.

One reviewer independently made these judgements, with another reviewer checking them. Disagreements between reviewers on study selection, data extraction or quality assessments were resolved by discussion; and where consensus could not be reached, they were resolved through referral to a third reviewer.

## Data synthesis

The studies were grouped as follows:

- Management of growth failure/faltering in infants

- Management of moderate wasting

- Management of severe wasting and/or bilateral pitting oedema

- Management of moderate wasting and severe wasting and/or bilateral pitting oedema together

Within these main groups, studies were sub-divided into sub-groups according to the type of management and setting (e.g., initiation of treatment in a community setting, initiation of treatment in outpatient settings, etc.). When reporting our results within these subgroups, we distinguished between evidence from studies of children 6 to 59 months versus 0 to 59 months; or studies of infants <6 months versus <12 months. This was in order to align our work with

the evidence requirements of the WHO guidelines for the prevention and management of acute malnutrition in infants and children under 5 years.

Where appropriate, we summarized quantitative outcomes descriptively using means, medians and ranges according to the perspectives adopted by the included studies. This descriptive analysis was conducted using Microsoft Excel. We also used tables, graphs and figures to present and visualize the data. There was significant heterogeneity with regards to interventions, settings, resource use, costs and costing methods such that pooled estimates would not generate robust or meaningful results [19, 20]. We, therefore, performed narrative syntheses to summarize the study results within the sub-groups.

To allow for comparability at an international level, all costs were converted to 2020 US dollars using purchasing power parity (PPP) exchange rates which account for variations between countries in the costs of goods and services [21, 22]. 2021 PPP exchange rates were not available at the time of data extraction. Conversion to 2020 USD was done after allowing for inflation using country-specific consumer price indexes [23].

### Presentation to the WHO Guideline Development Group (GDG)

The methods and results of the systematic review were shared with the WHO Prevention and Management of Wasting and Nutritional Oedema GDG for validation. The group comprises interdisciplinary international experts in child nutrition. We also presented our findings to this group during their GDG meeting on 21 to 24 March 2023. During the meeting, the GDG members discussed the review findings and asked any clarification questions. We revised our review according to the feedback provided where necessary.

## Results

### Search results

A total of 16,709 records were identified (Fig 1). The titles and abstracts of 9663 records were screened after removing duplicates. Of the 153 full texts sought for retrieval, seven either did not have a full text [24] or could not be retrieved even after contacting the authors [25–30]. 146 full texts were retrieved and screened, and 115 records were excluded [31–145]. Thus, 31 reports [146–176] representing 31 unique studies were included in the review. Details of excluded studies are provided in S5 File.

### General study characteristics

General study characteristics are shown in Table 3. Below is a narrative summary of the following characteristics: WHO region in which each study was conducted, study population, intervention, setting, outcomes explored, type of economic evaluation and cost items and costing methods. The study-level definitions for moderate wasting, severe wasting and/or bilateral pitting oedema and growth failure/faltering are provided as S6 File.

**WHO region.** Sixty-one percent (19/31) of the studies included at least one country in the African Region (Table 3) [147, 150, 152–155, 157, 158, 160–163, 167, 169, 171–174, 176]. Twenty-three percent (7/31) of studies included countries in the South-East Asian region [148, 149, 151, 156, 164–166]. Thirteen percent (4/31) of studies included countries in the East Mediterranean [146, 157, 168, 170], and 6% (2/31) of studies included countries in the American region [159, 175]. Over 90% of the countries were LMICs. South Africa was the only upper-middle income country, while Chile and USA were the only high-income countries.

**Study population.** Twenty studies (65%) were on infants and children with severe wasting and/or bilateral pitting oedema [146–151, 154–157, 160–162, 164, 170–174, 176]. Three studies

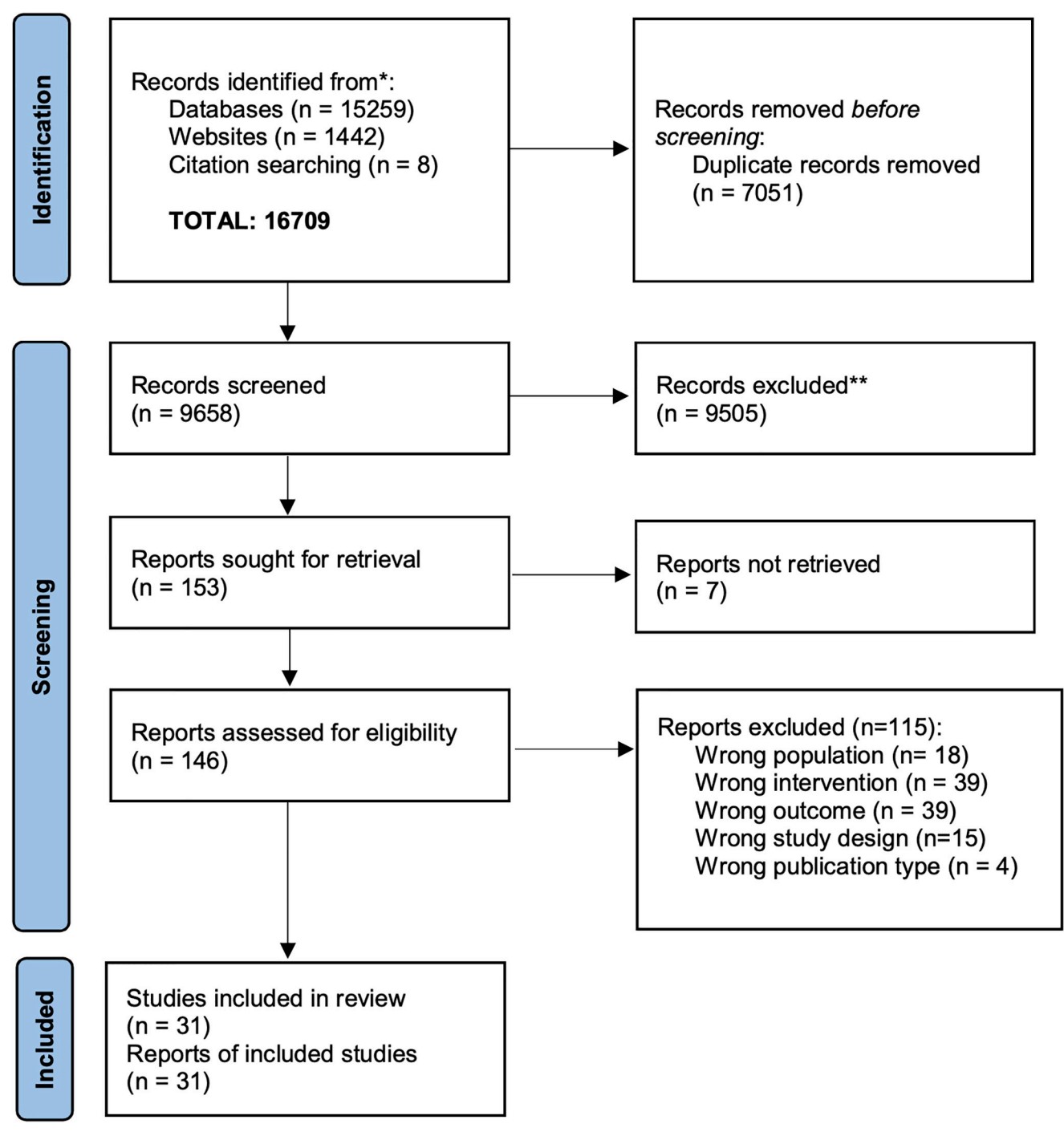

**Fig 1. Flow of studies through the review process.**

(10%) were among infants and children with moderate wasting [165, 168, 169]. There was only one study on growth faltering or failure (3%) [159]. The remaining 7 studies (23%) focused on both moderate and severe wasting and/or bilateral pitting oedema [152, 153, 158, 163, 166, 167, 175]. The age group covered by each study is shown in Table 3.

**Table 3. General characteristics of included studies.**

| Author, year | Country, WHO region | Population | Intervention | Setting (rural/urban), level of care/treatment setting | Outcomes | Study design | Type of economic evaluation | Cost data collection method | Cost perspective | Sample size | Reporting quality based on CHEERS checklist and % score+ |
|---|---|---|---|---|---|---|---|---|---|---|---|
| Akram (2016) [146] | Pakistan; Eastern Mediterranean | Severe wasting and/or bilateral pitting oedema; 6–23 months | Initiation of treatment in a community setting | Rural; home | Cost per child rehabilitated | Retrospective analysis | Cost analysis | NR | Provider | 123 | Low; 39% |
| Ali (2017) [147] | Nigeria; Africa | Severe wasting and/or bilateral pitting oedema; <60 months#^ | Initiation of treatment in outpatient settings | Not reported (NR); Outpatient therapeutic centre | Cost per: child, DALY averted, & life saved | Decision analytic model | Cost-effectiveness analysis (CEA) | Bottom-up | Societal | NR | Good; 79% |
| Ashraf (2019) [148] | Bangladesh; South-East Asia | Severe wasting and/or bilateral pitting oedema; 2–59 months#^ | Initiation of treatment in outpatient settings | Urban; day clinic | Cost per child treated | RCT | Cost analysis | Ingredients | Societal | 235 | Moderate; 55% |
| | | | Initiation of treatment in an inpatient setting | Urban; Inpatient hospital care | | | | | | 235 | |
| Ashworth (1997) [149] | Bangladesh; South-East Asia | Severe wasting and/or bilateral pitting oedema; 12–60 months | Initiation of treatment in an inpatient setting | Urban; children's nutrition unit | Cost per child rehabilitated | RCT | Cost analysis | Bottom-up & ingredients | Societal | 173 | Moderate; 66% |
| | | | Transfer from outpatient treatment at a health facility day care centre to domiciliary care | Urban; household care | | | | | | 130 | |
| | | | Initiation of treatment in outpatient settings | Urban; health facility day care centre | | | | | | 134 | |
| Bachmann (2009) [150] | Zambia; Africa | Severe wasting and/or bilateral pitting oedema; <60 months#^ | Initiation of treatment in outpatient settings | Urban; PHC | Cost per: child, death averted & DALY averted | Decision analytic modelling | CEA | Ingredients | Provider | 2523 | Good; 95% |

(Continued)

**Table 3.** (Continued)

| Author, year | Country, WHO region | Population | Intervention | Setting (rural/urban), level of care/treatment setting | Outcomes | Study design | Type of economic evaluation | Cost data collection method | Cost perspective | Sample size | Reporting quality based on CHEERS checklist and % score[+] |
|---|---|---|---|---|---|---|---|---|---|---|---|
| Bai (1972) [151] | India; South-East Asia | Severe wasting and/or bilateral pitting oedema; <60 months[#^] | Initiation of treatment in outpatient settings | Rural; health centre | Cost per child | Prospective cohort | Cost analysis | Bottom-up | NR | 25 | Low; 44% |
| Bailey (2020) [152] | Kenya, South Sudan; Africa | Moderate wasting and Severe wasting and/or bilateral pitting oedema; 6–59 months | Initiation of treatment in outpatient settings | Rural & urban; PHC | Cost per child recovered & incremental cost per child recovered | Cluster-randomised controlled non-inferiority trial | CEA | Ingredients & bottom-up | Societal | 2071; 2039 | Good; 84% |
| Chapko (1994) [153] | Niger; Africa | Moderate wasting and Severe wasting and/or bilateral pitting oedema; 5–28 months | Transfer from inpatient to outpatient | Urban; ambulatory rehabilitation centre | Cost per child treated | RCT | Cost analysis | Bottom-up | Provider | 47 | Moderate; 58% |
| | | | Transfer from an inpatient to another inpatient facility | Urban; hospital | | | | | | 53 | |
| Fotso (2019) [154] | Ethiopia; Africa | Severe wasting and/or bilateral pitting oedema; <60 months[#^] | Initiation of treatment in outpatient settings | NR; Health centres and health posts | Cost per: child cured, death averted & DALY averted | Prospective cohort | CEA | Bottom-up | Societal | 891; 1,286 | Good; 78% |
| Frankel (2015) [155] | Nigeria; Africa | Severe wasting and/or bilateral pitting oedema; NR months | Initiation of treatment in outpatient settings | NR; PHC | Cost per: child cured, death averted, & DALY averted | Prospective cohort and CEA model | CEA | Bottom up | Societal | NR | Good; 91% |
| Garg (2018) [156] | India; South-East Asia | Severe wasting and/or bilateral pitting oedema; 6–59 months | Initiation of treatment in a community setting | Urban; home | Cost per: child treated & child recovered | RCT | Cost analysis | Bottom-up | Provider | 124; 123 | Moderate; 68% |

(*Continued*)

**Table 3.** (Continued)

| Author, year | Country, WHO region | Population | Intervention | Setting (rural/urban), level of care/treatment setting | Outcomes | Study design | Type of economic evaluation | Cost data collection method | Cost perspective | Sample size | Reporting quality based on CHEERS checklist and % score[+] |
|---|---|---|---|---|---|---|---|---|---|---|---|
| Gomez (1983) [175] | Chile; Americas | Moderate wasting and Severe wasting and/or bilateral pitting oedema; 0–23, 24–71 & 0–71 months[#^] | Treatment initiation in the outpatient | Urban; Outpatient | Cost per: child recovered/day & child/day | Retrospective Cohort | Cost analysis | Top down | Societal | 745 | Moderate; 62% |
| | | | Treatment initiation in the community | Urban; Kindergarten | | | | | | 420 | |
| International Rescue Committee (IRC) (2016) [157] | Mali; Africa | Severe wasting and/or bilateral pitting oedema; <60 months[#^] | Initiation of treatment in outpatient settings | NR; PHC | Cost per child treated | Retrospective cohort | Cost analysis | NS | Provider | 2838; 2874; 6324 | Moderate; 50% |
| | Niger; Africa | | | | | | | | | 4976 | |
| | Kenya; Africa | | | | | | | | | 4,000*; 4,250* | |
| | Yemen; Eastern Mediterranean | | | | | | | | | 400*; 250* | |
| Isanaka (2017) [160] | Niger; Africa | Severe wasting and/or bilateral pitting oedema; 6–59 months | Initiation of treatment in a community/inpatient/outpatient settings | Rural; hospital | Cost per child treated | Cross-sectional | Cost analysis | Top-down | Provider | 6,903 | Moderate; 74% |
| | | | | Rural; community | | | | | | 13,395 | |
| | | | | Rural; hospital & community | | | | | | 20,298 | |
| Isanaka (2019) [158] | Mali; Africa | Moderate wasting and Severe wasting and/or bilateral pitting oedema; 6–35 months | Initiation of treatment in outpatient settings | Rural; community health centre | Cost per child identified, incremental cost per: death & DALY averted | Decision tree: Cluster-randomised trial | CEA | Ingredients & bottom-up | Provider | 1,766 | Good; 87% |
| Karniski (1986) [159] | USA; Americas | Growth faltering; <12 months[^] | Transfer from inpatient to community settings | Urban; medical placement home | Cost per child treated | Retrospective cohort | Cost analysis | Ingredients and top-down | Provider | 17 | Moderate; 55% |
| | | | Initiation of treatment in an inpatient setting | Urban; hospital | | | | | | 18 | |

(Continued)

**Table 3.** (Continued)

| Author, year | Country, WHO region | Population | Intervention | Setting (rural/ urban), level of care/treatment setting | Outcomes | Study design | Type of economic evaluation | Cost data collection method | Cost perspective | Sample size | Reporting quality based on CHEERS checklist and % score[+] |
|---|---|---|---|---|---|---|---|---|---|---|---|
| Masiiwa (2013) [161] | Zimbabwe; Africa | Severe wasting and/or bilateral pitting oedema; 0–59 months[#^] | Referral to treatment in an inpatient setting | Urban; hospital | Cost per household | Cross sectional study | Cost analysis | Bottom-up | Household | 142 | Good; 84% |
| N'Diaye (2020) [162] | Burkina Faso; Africa | Severe wasting and/or bilateral pitting oedema; 6–59 months | Initiation of treatment in outpatient settings | Urban & non-urban; PHC | Cost per child treated | Decision analytic model (Part of a clinical trial) | Cost minimisation analysis | Top-down & bottom-up | Societal | 399; 399 | Good; 92% |
| Nkonki (2017) [163] | South Africa; Africa | Moderate wasting; 0–59 months[#^] | Initiation of treatment in a community setting | NR; community | Total cost | Deterministic mathematical model | Cost analysis | Ingredients | Provider | NR | Moderate; 63% |
| | | Severe wasting and/or bilateral pitting oedema; 0–59 months[#^] | | | | | | | | | |
| Puette (2013) [164] | Bangladesh; South-East Asia | Severe wasting and/or bilateral pitting oedema; 6–36 months | Referral to treatment in an inpatient setting | Rural; Upazila health complex | Cost per: child treated, recovered, death averted, & DALY averted | Cost model | Cost analysis | Top-down & bottom-up | Societal | 633 | Good; 80% |
| | | | Initiation of treatment in outpatient settings | Rural; community | | | | | | 724 | |
| Purwestri (2012) [165] | Indonesia; South-East Asia | Moderate & mild wasting; 6–59 months | Initiation of treatment in a community setting | Semi-urban & rural; community | Cost per: child & child reaching discharge criterion | Cohort study | Cost analysis | Top-down & bottom-up | Societal | 103;101 | Moderate; 72% |
| Reed (2012a) [168] | Pakistan; Eastern Mediterranean | Moderate wasting; 6–59 months | Initiation of treatment in outpatient settings | Rural & urban; NS | Cost per beneficiary | Retrospective study | Cost analysis | Top-down | Provider | 57,946 | Moderate; 56% |
| | | | | Rural & urban; hospital | | | | | | NS | |
| | | | | Rural & urban; outpatient centres | | | | | | 12,701 | |

*(Continued)*

**Table 3.** (Continued)

| Author, year | Country, WHO region | Population | Intervention | Setting (rural/urban), level of care/treatment setting | Outcomes | Study design | Type of economic evaluation | Cost data collection method | Cost perspective | Sample size | Reporting quality based on CHEERS checklist and % score[+] |
|---|---|---|---|---|---|---|---|---|---|---|---|
| Reed (2012b) [167] | Kenya; Africa | Severe wasting and/or bilateral pitting oedema; 6–59 months | Initiation of treatment in outpatient settings | Arid, semi-arid lands and urban; PHC | Cost per child treated | Cross-sectional study | Cost analysis | Top-down | Provider | 13,501 | Moderate; 56% |
| | | 6–59 months | Referral to treatment in an inpatient setting | Arid, semi-arid lands and urban; Hospital | | | | | | 990 | |
| | | Moderate wasting; 6–59 months | Initiation of treatment in a community setting | Arid, semi-arid lands and urban; Community based care | | | | | | 44,148 | |
| Reed (2012c) [166] | Nepal; South-East Asia | Severe wasting and/or bilateral pitting oedema; 6–59 months | Initiation of treatment in outpatient settings | NR; PHC | Cost per child treated | Cross-sectional study | Cost analysis | Top-down | Provider | 7,548 | Moderate; 56% |
| | | Moderate wasting; 6–59 months | Initiation of treatment in a community setting | NR; Community based care | | | | | | 40,769 | |
| Rogers (2017) [169] | Malawi; Africa | Moderate wasting; 6–59 months | Initiation of treatment in a community setting | Rural; home | Cost per: treated beneficiary & additional caregiver meeting or exceeding target | Cross-sectional study | CEA | Top-down & bottom-up | Societal | 196; 192; 196 | Moderate; 63% |
| Rogers (2018) [171] | Mali; Africa | Severe wasting and/or bilateral pitting oedema; 6–59 months | Initiation of treatment in a community setting | Rural; community | Cost per: child treated & child recovered | Clinical cohort trial | CEA | Top-down & bottom-up | Societal | 617 | Good; 76% |
| Rogers (2019) [170] | Pakistan; Eastern Mediterranean | Severe wasting and/or bilateral pitting oedema; 6–59 months | Initiation of treatment in outpatient settings | NR; PHC | Cost per child recovered & incremental cost per additional child recovered | Cost model (Part of a clinical trial) | CEA | Top-down & bottom-up | Provider (institutions); Provider (government); Community | 393 | Good; 82% |
| | | 6–59 months | Initiation of treatment in a community setting | NR; community | | | | | | 425 | |

(*Continued*)

**Table 3.** (Continued)

| Author, year | Country, WHO region | Population | Intervention | Setting (rural/urban), level of care/treatment setting | Outcomes | Study design | Type of economic evaluation | Cost data collection method | Cost perspective | Sample size | Reporting quality based on CHEERS checklist and % score[+] |
|---|---|---|---|---|---|---|---|---|---|---|---|
| Tekeste (2012) [172] | Ethiopia; Africa | Severe wasting and/or bilateral pitting oedema; 6–59 months | Initiation of treatment in a community setting | Rural; community | Cost per: child cured & child treated | Retrospective comparative study | CEA | Top-down | Societal | 157 | Good; 82% |
| | | | Initiation of treatment in outpatient settings | Rural; therapeutic feeding centre | | | | | | 149 | |
| UNICEF (2012) [176] | Chad; Africa | Severe wasting and/or bilateral pitting oedema; 0–59 months[#][^] | Treatment initiation in the outpatient/inpatient | Rural; Outpatient | Cost per child | Descriptive analysis | Cost analysis | Top down and ingredients-based | Provider | NR | Low; 47% |
| Wilford (2011) [173] | Malawi; Africa | Severe wasting and/or bilateral pitting oedema; <60 months[#][^] | Initiation of treatment in a community/inpatient/outpatient settings | NR; hospital/PHC/community | Cost per child treated & per DALY averted | Cross-sectional study | CEA | Top-down | Provider | 3,577 | Good; 84% |
| Wilunda (2021) [174] | Tanzania; Africa | Severe wasting and/or bilateral pitting oedema; 6–59 months | Initiation of treatment in outpatient settings | Rural; PHC | Cost per: child treated & cured & incremental cost per additional child cured | Non-inferiority quasi-experimental study (Part of a clinical trial) | CEA | Bottom-up | Provider | 154 | Good; 82% |
| | | | Initiation of treatment in a community setting | Rural; home | | | | | | 210 | |

^No subgroup analysis for those aged <6 months

#No subgroup analysis for those aged 6–59 months

+Excludes items 23 & 24 of the CHEERS checklist

**Interventions and settings.** Nineteen studies (61%) evaluated the initiation of treatment in outpatient settings [147, 149–152, 154, 155, 157, 158, 162, 164, 166–168, 170, 172, 174–176] and 13 (42%) in community settings [146, 148, 156, 163, 165–167, 169–172, 174, 175]. Three studies (10%) evaluated referral from outpatients or community settings to inpatient settings [161, 164, 167], and another 3 (10%) were on the initiation of treatment in a mixture of settings [160, 173, 176]. There was one study (3%) each for transfer from outpatients to community settings [149], inpatients to another inpatient setting [153], inpatient to outpatient settings [153], and inpatient to community settings [159].

**Outcomes.** As shown in Table 3 the outcomes that were reported varied widely. These outcomes were often not clearly defined. When provided, definitions varied widely between studies for the same outcome. For example, the definition for cost per child treated varied from the cost per child admitted for treatment, regardless of the outcome [167], cost per child discharged regardless of the outcome [170, 173], through to cost per child admitted for treatment and successfully recovered [172]. For the purposes of this review, we assumed that "child recovered", "child cured" or "child rehabilitated" referred to the cost per child admitted for treatment and successfully recovered; whilst "child treated", "child covered", "child identified", "treated beneficiary", "child", and "beneficiary" referred to the cost per child admitted for treatment, regardless of the outcome.

**Type of economic evaluation.** Eighteen (58%) studies were cost or cost-efficiency analyses (Table 3; see S2 File for definitions) [146, 148, 149, 151, 153, 156, 157, 159–161, 163, 164, 166–168, 175, 176], and 12 studies (39%) were cost-effectiveness analyses [147, 150, 152, 154, 155, 158, 169–174]. N'Diaye and colleagues carried out a cost-minimisation analysis to identify the option that results in the lowest costs [162].

**Cost items and costing methods.** The items that were costed and the costing method used by each study are shown in Table 4. The type of costs and costing methods are also defined in S2 File.

## Management of growth failure/faltering in infants <12 months of age

There was one study on infants aged <12 months conducted in the USA which found that transfer from inpatient to community treatment in a medical placement home was $1003 more expensive per child treated than treatment in an inpatient setting from a provider perspective ($6,776 versus $5,773; Table 5, S1 Table) [159]. The study included those diagnosed with non-organic failure to thrive and excluded those with any indication of organic aetiology for the failure to thrive. The specific treatments provided for growth failure/faltering were not clearly specified, but the analysed costs included the costs of physicians, laboratory tests, radiology, medication and room charges. We did not find any study reporting on infants <6 months of age. No studies were found for the other treatment setting decisions and outcomes of interest in this group.

## Management of moderate wasting in infants and children <60 months of age

Studies identified in this group reported on costs, cost-efficiency and/or cost-effectiveness of initiating treatment in community or outpatient settings as summarised below.

**Initiation of treatment in community settings.** The costs or cost-efficiency of initiating treatment for moderate wasting in community settings were reported by five studies: four in those aged 6 to 59 months [165–167, 169] and one in those aged 0 to 59 months [163]. The cost per child treated varied widely: from the provider perspective this was from $18 to $934 [165–167], and from a societal perspective it was from $237 to $1,380 (Table 6; S2 Table) [165,

**Table 4. Costs and data collection methods.**

| Author, Year | Reference year of costs[+] | Costing method | Personnel | Training | Administrative | Capital | Diagnostics | Medication | Transport (provider) | Transport (household) | Home food | Specially formulated foods | Productivity loss | Work for care giver | Other |
|---|---|---|---|---|---|---|---|---|---|---|---|---|---|---|---|
| Akram (2016) [146] | 2012 | Incremental financial cost, NR approach | x | | | | | | x | | | x | | | Parental counselling |
| Ali (2017) [147] | (2016) | Incremental economic cost, bottom-up approach | x | | x | | | x | | | | x | | | Community volunteer |
| Ashraf (2019) [148] | (2018) | Full economic cost, ingredients approach | x | | | x | x | x | x | x | x | x | x | | Supportive care & hotel |
| Ashworth (1997) [149] | (1996) | Full financial cost, bottom-up & ingredients approach | x | | | x | x | x | | x | x | x | x | | Other parental costs |
| Bachmann (2009) [150] | 2008 | Incremental financial cost, ingredients approach | | x | | | | | x | | | x | | | Hospitalisation, community mobilisation & health centre visits |
| Bai (1972) [151] | (1971) | Incremental economic cost, bottom-up approach | | | | | | | | | | x | x | | |
| Bailey (2020) [152] | 2017 | Incremental economic cost, ingredients & bottom-up approach | x | | | x | | x | x | | | x | | | Outreach |
| Chapko (1994) [153] | (1993) | Full financial cost, bottom-up approach | x | | | x | x | x | | | x | x | | | |
| Fotso (2019) [154] | (2018) | Incremental economic cost, bottom-up approach | x | x | x | | x | x | x | x | | x | x | | |
| Frankel (2015) [155] | (2014) | Full economic cost, bottom-up approach | x | x | x | x | x | x | x | | | x | x | | |
| Garg (2018) [156] | 2014 | Incremental financial cost, bottom-up approach | x | x | x | x | x | x | x | | | x | | | Peer support |
| Gomez (1983) [175] | 1979 | Incremental Economic cost, top-down approach | x | | | x | | | | | | x | | x | Distribution and other |

*(Continued)*

**Table 4.** (Continued)

| Author, Year | Reference year of costs+ | Costing method | Personnel | Training | Administrative | Capital | Diagnostics | Medication | Transport (provider) | Transport (household) | Home food | Specially formulated foods | Productivity loss | Work for care giver | Other |
|---|---|---|---|---|---|---|---|---|---|---|---|---|---|---|---|
| IRC (2016) [157] | (2015) | Full financial cost, NS approach; Incremental financial cost, NS approach | x | | x | x | | x | | | | x | | | |
| Isanaka (2017) [160] | 2015 | Full financial cost, top-down approach | x | | x | x | x | x | x | | | x | | | Non-medical equipment &supplies |
| Isanaka (2019) [158] | 2015 | Full financial cost, ingredients & bottom-up approach | x | | x | x | | | x | | | x | | | Medical supplies & materials |
| Karniski (1986) [159] | (1985) | Incremental financial cost, ingredients and top-down approach | x | | x | | | | x | | | x | | | Hospital charges & overheads |
| Masiiwa (2013) [161] | (2012) | Incremental economic cost, bottom-up approach | x | | | | | x | | x | x | | x | | |
| N'Diaye (2020) [162] | 2017 | Incremental economic cost, top-down & bottom-up approach | x | | | | | x | | | | x | x | | Material i.e., thermometers, height boards, weight scales, & transportation boxes |
| Nkonki (2017) [163] | 2015 | Incremental financial cost, ingredients approach | x | | | x | | x | | | | | | | Other recurrent costs |
| Puette (2013) [164] | 2010 | Incremental economic cost, top-down & bottom-up approach | x | x | | x | | x | | x | x | x | | | Other household costs |
| Purwestri (2012) [165] | 2007 | Incremental economic cost, top-down & bottom-up approach | x | | | x | x | | x | | | x | x | | Volunteer incentives |
| Reed (2012a) [168] | (2011) | Incremental financial cost, top-down approach | x | | | x | | | | | | | | | Utilities, medical equipment& service charges |

(*Continued*)

**Table 4.** (Continued)

| Author, Year | Reference year of costs+ | Costing method | Personnel | Training | Administrative | Capital | Diagnostics | Medication | Transport (provider) | Transport (household) | Home food | Specially formulated foods | Productivity loss | Work for care giver | Other |
|---|---|---|---|---|---|---|---|---|---|---|---|---|---|---|---|
| Reed (2012b) [167] | (2011) | Incremental financial cost, top-down approach | x | | | | | x | | | | x | | | |
| Reed (2012c) [166] | (2011) | Incremental financial cost, top-down approach | x | | | | | | | | | x | | | |
| Rogers (2017) [169] | 2014 | Incremental economic cost, top-down & bottom-up approach | x | x | x | | | | x | | | x | x | | Warehousing |
| Rogers (2018) [171] | 2016 | Incremental financial cost, top-down & bottom-up approach | x | x | x | | | | x | x | | x | x | | Community level rent, utilities & rent |
| Rogers (2019) [170] | (2018) | Incremental economic cost, top-down & bottom-up approach | x | x | x | | | | x | | | x | x | | Community level rent, utilities & rent |
| Tekeste (2012) [172] | 2006 | Full economic cost, top-down approach | x | | | x | | x | | x | x | x | x | x | Caregiver's food |
| UNICEF (2012) [176] | 2010 | Incremental Financial cost, top down and ingredients-based approach | x | x | x | x | | x | x | | | x | | | Other |
| Wilford (2011) [173] | 2007 | Incremental costs financial cost, top-down approach | x | x | x | x | | x | x | | | x | | | Inpatient costs |
| Wilunda (2021) [174] | 2019 | Incremental financial cost, bottom-up approach | x | x | x | x | | | | | x | | | | Sensitization & mobilisation |

**Table 5. Costs and cost-effectiveness of transfer and referral models of care.**

| | Community to outpatient | Community/ outpatient to inpatient | Inpatient to outpatient/ community | Outpatient to community |
|---|---|---|---|---|
| **What are the associated costs** | | | | |
| Growth failure/faltering in infants <12 months of age | No studies identified | No studies identified | *Age <12 months (to community)* **Provider costs** $6,776 per child treated | No studies identified |
| Moderate wasting among infant and children <60 months of age | No studies identified | No studies identified | No studies identified | No studies identified |
| Severe wasting and/or pitting oedema among infant and children <60 months of age | No studies identified | *Age 6 to 59 months* **Provider costs**: $298–$714 per child treated **Societal costs**: $5,465 per child treated $37,204 per child recovered *Age < 60 months* **Household costs**: $84 per household | No studies identified | *Age 12 to 60 months* **Provider costs**: $163 per child recovered **Parentals costs**: $52 per child recovered |
| Moderate wasting and severe wasting and/or bilateral pitting oedema together among infants and children <60 months of age | No studies identified | No studies identified | *Age 5 to 28 months (to outpatient)* **Provider costs**: $98 per child treated | No studies identified |
| **What is the cost-effectiveness** | | | | |
| Growth failure/faltering in infants <12 months of age | No studies identified | No studies identified | No studies identified | No studies identified |
| Moderate wasting among infant and children <60 months of age | No studies identified | No studies identified | No studies identified | No studies identified |
| Severe wasting and/or pitting oedema among infant and children <60 months of age | No studies identified | *Age 6 to 59 months* **Societal perspective**: $5465 per DALY averted (compared to no treatment) | No studies identified | No studies identified |
| Moderate wasting and severe wasting and/or bilateral pitting oedema together among infants and children <60 months of age | No studies identified | No studies identified | No studies identified | No studies identified |

169]. The treatments provided varied from the provision of counselling, which seemed to incur the lowest cost, to the provision of supplementary food (e.g., $18 for counselling [166] versus $943 for weekly supplementary food distribution from the provider perspective [165]).

In the counselling support programme in Nepal the cost per child treated was $18 from a provider's perspective [166]. Supplementary foods were only provided in food security emergencies [166]. In Kenya where the cost per child treated from a provider's perspective was $201, ready-to-use therapeutic food (RUTF) was provided [167]. In Indonesia, Purwestri et al reported costs per child recovered of $502 and $822 for daily distribution of supplementation food in a semi-urban community from the provider and societal perspectives, respectively [165]. In a rural community, the costs per child recovered were $467 and $604 for weekly

**Table 6. Costs and cost-effectiveness of initiating treatment in community and outpatient settings.**

| | Community setting | Outpatient setting |
|---|---|---|
| **What are the associated costs** | | |
| Growth failure/faltering in infants <12 months of age | No studies identified | No studies identified |
| Moderate wasting among infant and children <60 months of age | *Age 6 to 59 months* <br> **Provider costs**: <br> $18 -$943 per child treated <br> $467 & $502 per child recovered <br> **Societal costs**: <br> $237 -$1,380 per child treated <br> $604 & $822 per child recovered <br> *Age <60 months* <br> **Total costs**: <br> Total provider costs of nearly $6 million | *Age 6 to 59 months* <br> **Provider costs**: <br> $118 -$277 per child treated |
| Severe wasting and/or pitting oedema among infant and children <60 months of age | *Age 6 to 59 months* <br> *Initiation at home* <br> **Provider costs**: <br> $1,779 -$2,047 per child treated <br> $13 per child surveyed <br> $126 per child recovered <br> *Initiation through health workers working in proximity to health facilities or through community health workers* <br> **Provider costs**: <br> $202 to $799 per child treated <br> $270 to $1051 per child recovered <br> **Societal costs**: <br> $671–$757 per child treated <br> $732–$817 per child recovered <br> **Community costs**: <br> $64 per child treated <br> *Age <60 months* <br> **Total costs**: <br> Total provider costs of nearly $6 million | *Age 6 to 59 months* <br> **Provider costs**: <br> $103–$1267 per child treated <br> $239 to 1369 per child recovered <br> **Societal costs**: <br> $295 to 1597 per child treated <br> $1419 to 1796 per child recovered <br> **Community costs**: <br> $117 per child treated <br> $141 per child recovered <br> **Parental costs**: <br> $25 per child recovered <br> *Age < 60 months* <br> **Provider costs**: <br> $67 to $3374 per child treated <br> **Societal costs**: <br> $76 to $529 per child treated <br> $438 to 1129 per child recovered <br> **Household costs**: <br> $55 per child treated |
| Moderate wasting and severe wasting and/or bilateral pitting oedema together among infants and children <60 months of age | **Societal costs**: <br> Cost per child recovered per day of $15 for children and infants aged 0 to 23 months, and $ 21 for those aged 24 to 71 months | **Societal costs**: <br> Cost per child recovered per day of $6 for children and infants aged 0 to 23 months, and $11 for those aged 24 to 71 months |
| **What is the cost-effectiveness** | | |
| Growth failure/faltering in infants <12 months of age | No studies identified | No studies identified |
| Moderate wasting among infant and children <60 months of age | No studies identified | No studies identified |

(*Continued*)

**Table 6.** (Continued)

|  | Community setting | Outpatient setting |
|---|---|---|
| Severe wasting and/or pitting oedema among infant and children <60 months of age | *Age 6 to 59 months* <br> **Provider perspective**: $391 per additional child treated, $381 per additional child cured; Outpatient care as reference | *Age 6 to 59 months* <br> **Provider perspective**: <br> $1483 per child recovered; compared to community treatment by lady health workers <br> **Societal perspective**: <br> $106 per DALY averted compared to no treatment <br> $3534 per death averted compared to no treatment <br> *Age < 60 months* <br> **Provider perspective**: <br> $20 to $145 per DALY averted; compared to doing nothing or no programme implementation scenarios <br> **Societal perspective**: <br> $68 to $161 per DALY averted; compared to doing nothing or no programme implementation scenarios |
| Moderate wasting and severe wasting and/or bilateral pitting oedema together among infants and children <60 months of age | No studies identified | No studies identified |

distribution of supplementation food from the provider and societal perspectives, respectively [165].

In South Africa, Nkonki et al reported total provider costs of nearly $6 million for a multi-component intervention package that included the provision of supplementation food, maternal education and provision of oral rehydration solution for diarrhoea and oral antibiotics for suspected respiratory infections [163]. This study did not report the sample size.

In Malawi, Rogers et al [169] found that when compared to a control of monthly rations of 1 litre oil and 8kg corn-soy blend (CSB) and social and behaviour change communication (SBCC), an increased oil allocation (to 2.6 litres of oil for the 8 kg of CSB), either in bulk or four packages, plus enhanced SBCC were cost-effective at $249 and $396 per additional caregiver meeting or exceeding the recommended target of 30g oil:100g CSB respectively (S3 Table).

**Initiation of treatment in outpatient settings.** From the provider's perspective, the costs per child treated for the initiation of treatment in outpatient settings for infants and children within the age group 6 to 59 months ranged between $118 and $277 from two studies [158, 168] (Table 6; S2 Table). The programme in Pakistan resulting in a cost per child treated of $118 used ready-to-use supplementary food (RUSF), RUTF and fortified blended foods [168]. From the study by Isanaka et al in Mali, the cost per child treated ranged from $247 to $277 and did not seem to vary much with the different types of foods used, i.e., RUSF; a specially formulated corn–soy blend (CSB); Misola, a locally produced, micronutrient-fortified, cereal–legume blend (MI); or locally milled flour (LMF) [158]. Nevertheless, out of the four dietary supplements, moderate wasting treatment with RUSF was the most cost-effective across a wide range of cost-effectiveness thresholds, including WHO's gross domestic product (GDP) per capita threshold, with incremental cost-effectiveness ratios of $963 per DALY averted and $27,246 per death averted compared to no treatment of moderate wasting (S3 Table) [158]. Treatment using the other three dietary supplements were dominated (S3 Table) [158].

Unfortunately, there were no head-to-head comparisons between settings to enable judgements on whether one setting resulted in less costs or was more cost-efficient or cost-effective than the other.

## Management of severe wasting and/or bilateral pitting oedema in infants and children <60 months of age

Studies in this group covered initiation of treatment in community settings; initiation of treatment in outpatient settings; referral to treatment in an inpatient setting; and transfer from outpatient treatment to treatment in a community setting. Three studies also reported on initiating treatment in a mixture of settings.

**Initiation of treatment in community settings.**   There were nine studies in total identified in this group. Three of them reported on the initiation of treatment within the home for infants and children within the age group of 6 to 59 months [146, 156, 174] (S4 Table). One of these three studies was conducted in India, and the costs to the government per child treated ranged from $1,779 to $2,047, depending on the type of RUTF used [156] (Table 6; S4 Table). Centrally produced RUTF (RUTF-C) incurred the lowest costs, followed by locally prepared RUTF (RUTF-L), with micronutrient-enriched (augmented), energy-dense, home-prepared food (A-HPF) incurring the highest costs. They also reported a cost of $13 per child surveyed within the community from a provider (government) perspective. Treatment was initiated by Accredited Social Health Activist (ASHA)-like workers (ALW) in a research setting. The second study, conducted in Pakistan, involved a team comprising a doctor, a lay health supervisor, a lady health worker and a project supervisor visiting malnourished children in their homes to initiate treatment [146]. The study reported a cost per child recovered of $126 from a provider perspective using a locally produced indigenous high-density diet (HDD) plus 'Baby Active', a micronutrient powder sprinkled on any food the child consumed [146]. In Tanzania, Wilunda et al evaluated an intervention where treatment was at home using RUTF (type not specified), with the dosage based on a child's body weight [174]. Children were followed up through community health worker weekly home visits [174]. The reports costs were $426 per child treated and $470 per child recovered from a provider perspective.

The remaining six studies involved initiation of treatment within community settings either by health workers working in proximity to health facilities (Niger) [160], or by community health workers (covering Bangladesh, Pakistan, Mali, South Africa and Ethiopia) [163, 164, 170–172]. In certain cases, non-governmental organisations also provided outpatient care to complement these initiatives [170]. From the five that were among infants and children within the age group of 6 to 59 months, the cost per child treated ranged from $202 to $799 from the provider perspective, and from $671 to $757 from a societal perspective (Table 6; S4 Table). The cost per child recovered ranged from $270 to $1,051 from the provider perspective, and from $732 to $817 from a societal perspective. Rodgers et al also reported a cost per child treated of $64 and a cost per child recovered of $84 from the community perspective [170]. In South Africa, Nkoki et al's total programme costs for the treatment of children aged 0 to 59 months were about $13 million from the provider perspective but the population size was undocumented [163].

In the study by Isanaka et al in Niger where treatment initiation was by health workers, patients were provided with rations of RUTF, routine medical treatment and received weekly visits from health staff [160]. The remaining five studies involved community setting treatment initiation by community health workers. In Bangladesh, nutritional treatment included a weekly ration of RUTF, a single oral dose of folic acid (5 mg) and oral cotrimoxazole [164]. In Pakistan, there was provision of medical and nutritional treatment, and counselling on

nutrition and Infant and Young Child Feeding practices in the patient's home [170]. In Mali, Rodgers et al integrated the treatment of severe wasting into an existing Integrated Community Case Management programme [171]. In Tekeste et al's study in Ethiopia, treatment was in community-based therapeutic centres, and included medicines to treat complications and therapeutic foods [172]. Patients also received care from community volunteers and daily visits from a nurse [172]. In South Africa, the costs were estimated from a multicomponent intervention package that included the provision of supplementation food, maternal education, oral rehydration solution for diarrhoea and oral antibiotics for respiratory infections [163].

In terms of cost-effectiveness, Wilunda et al reported that initiation of treatment in community settings was cost-effective, with an estimated incremental costs per additional child treated or cured of $391 and $381, respectively, when compared to initiation of treatment in outpatient settings in Tanzania using WHO's GDP/capita threshold [174] (Table 6; S5 Table).

**Initiation of treatment in outpatient settings.** There were 17 studies included in this group (S4 Table). Initiation of treatment in outpatient settings was mostly in primary health clinics [150, 151, 154, 155, 157, 158, 162, 166–168, 170, 171, 174], with two at therapeutic feeding centres [147, 172] and two at day centres [148, 149]. The studies covered Bangladesh, Burkina Faso, Ethiopia, India, Kenya, Mali, Nepal, Niger, Nigeria, Pakistan, Tanzania, Yemen and Zambia.

Ten of the 17 studies were among children aged 6 to 59 months [149, 158, 162, 166–168, 170–172, 174]. For these 10 studies, the cost per child treated ranged from $103 to $1267 from a provider perspective, $205 to 1597 from a societal perspective, and $117 from a community perspective (Table 6; S4 Table). The cost per child recovered was between $239 and $1369 from a provider perspective; ranged from $1419 to $1796 from a societal perspective; approximately $141 from one study from a community perspective; and $25 for the parents.

The remaining seven studies were among children aged <60 months and did not include a subgroup analysis for those aged 6 to 59 months [147, 148, 150, 151, 154, 155, 157]. We were able to summarise costs for six of these. The cost per child treated ranged from $67 to $3374 from a provider perspective; $76 to 529 from a societal perspective, and approximately $55 from a household perspective (Table 6; S4 Table). The cost per child recovered ranged from $438 to $1129 from a societal perspective. The remaining study, Bai et al, reported a cost of $6,365 per child treated in India, but the perspective was not clear [151].

For most of the 17 studies under this group, care was provided by health workers, mostly nurses, with Lady Health Workers providing care in one study in Pakistan [168]. Most studies included the provision of RUTF, micronutrients such as folic acid or Vitamin A, broad-spectrum antibiotics and health education. Health education included encouragement to breastfeed, milk-based therapeutic diets or other modified diets (e.g., increasing the intake of pulses, nuts and eggs, rice-based meals etc) [148, 149, 151].

Six studies carried out cost-effectiveness analyses of initiation of treatment of severe wasting and/or bilateral pitting oedema in outpatient settings (S5 Table) [147, 150, 154, 155, 164, 170]. These studies reported cost per disability-adjusted life year (DALY) gained/averted [147, 150, 154, 155, 164], cost per death averted [150, 154, 155, 164], cost per life saved [147], and cost per additional child recovered [170] (Table 6; S5 Table). Only two of these studies were specific to the age group 6 to 59 months. One of these two, conducted in Bangladesh, reported that initiating treatment in outpatient settings was highly cost-effective when compared to no treatment, with a cost per DALY averted of $106 and a cost per death averted of $3534 from a societal perspective [164]. Inpatient treatment initiation costed much more by comparison, $5,465 per DALY averted and $185,787 per death averted when compared to no treatment from a societal perspective. The second study, conducted by Rogers et al in Pakistan, had a cost per additional child recovered of $1,483 for outpatient treatment when compared to

community treatment by lady health workers from a provider perspective [170]. The differences in costs and recovery rates between the two strategies were small, which resulted in uncertainty in terms of which strategy was the most cost-effective.

The remaining four studies included infants and children <60 months in Nigeria [147, 155], Ethiopia [154] and Zambia [150]. When compared to doing nothing or no programme implementation, initiation of treatment in an outpatient setting was considered to be highly cost-effective using country-specific WHO GDP per capita thresholds. Cost per DALY gained/averted were $20 [150] and $145 [147] from the provider perspective, and $68 [155] and $161 [147] from a societal perspective (Table 6; S5 Table). Fotso et al reported that incorporating a surge approach in Ethiopia to strengthen the health system's resilience against seasonal 'surges' in the demand for treatment of acute malnutrition, and a standard approach of delivering SAM treatment in outpatient were both highly cost-effective using the GDP per capita threshold [154]. However, the surge approach was reported as less cost effective with a cost per DALY averted of $70 (95%CI: $53–$91) when compared to the standard service which had a cost per DALY averted of $35 (95%CI: $27–$48). One study also estimated a cost per death averted from a provider perspective of $677 for initiating treatment in outpatient settings compared to doing nothing in Zambia [150], and this was $2,480 from a societal perspective from one study in Nigeria [155]. From the study by Fotso et al the cost per death averted were $4,973 (95%CI: $3,640–$7,089) with the surge approach, and $2,480 (95%CI: $1,891–$3,781) for the standard services. Ali et al also reported a cost per life saved of $5,376 from a provider perspective and $5,951 from a societal perspective for initiating treatment in outpatient settings when compared to no programme implementation (S5 Table) [147].

**Referral to treatment in an inpatient setting.**   There were four studies identified for this group. Two studies included an investigation of the costs of a model of care where children aged 6 to 59 months with severe wasting and/or bilateral pitting oedema and medical complications started their treatment in community and outpatient settings and were then referred for stabilisation in inpatient settings [167, 168]. The costs per child treated were $298 in Kenya [167] and $714 in Pakistan [168] from the provider perspective (Table 5; S4 Table). In the study by Puette et al in Bangladesh, a care model involving referral from trained community health workers to inpatient hospital treatment among 6- to 36-month-olds with severe wasting and/or bilateral pitting oedema resulted in costs of $5,465 per child treated and $37,204 per child recovered from a societal perspective [164]. This was not cost-effective from a societal perspective, at $5,465 per DALY averted when compared to no treatment (Table 5; S5 Table) [164]. Masiiwa had a cost per household of $84 for a model of care where children aged less than 60 months with severe wasting and/or bilateral pitting oedema were referred from community and outpatient primary health care clinic treatment to hospital inpatient clinic in Zimbabwe (Table 5; S4 Table) [161].

**Transfer from outpatient treatment to treatment in a community setting.**   In their study in Bangladesh, Ashworth and Khanum included a group of children, aged from 12 to 60 months, that were first treated for one week in the day care facility before sending them to be cared for at home [149]. During their care at home, they were visited by specially trained health care workers every week for one month, and then fortnightly, until reaching a weight-for-height that is 80% of the National Centre for Health Statistics median. The cost per child recovered was $163 from the provider perspective and $52 from the household perspective for this model of care (Table 5; S4 Table).

**Initiation of treatment in multiple settings.**   Three studies were identified for this group. In Niger, Isanaka et al reported a cost per child treated of $399 for a cohort where some children, aged 6 to 59 months, initiated treatment in hospital and some in community settings [160]. In the Chad UNICEF study among children aged 0 to 59 months, initiation of treatment

in outpatient or community settings resulted in a cost per child treated of $505 [176]. In the study by Wilford et al in Malawi among the 0 to 59 months age group, some initiated treatment in hospital, some in primary health care and some in community settings [173]. The cost per child treated was $445 when including standard health services, and $44 when the standard health service was excluded, from a provider perspective [173]. The cost-effectiveness analysis resulted in a cost per DALY averted of $110 when standard health services were included versus when they were not [173].

## Management of moderate wasting and severe wasting and/or bilateral pitting oedema together in infants and children <60 months of age

Studies identified for this group reported the cost and cost-effectiveness analysis for initiating treatment in outpatient settings and transfer from inpatient to outpatient/community settings.

**Initiation of treatment in outpatient settings.** There were two studies identified for this group. Bailey et al estimated the cost of initiation of treatment in outpatient settings for infants and children aged 6 to 59 months in Kenya [152]. They found that, from a societal perspective, using different standard protocols for severe wasting and/or bilateral pitting oedema and moderate wasting was $337 more expensive per child recovered compared to using the same combined protocol for both severe wasting and/or bilateral pitting oedema and moderate wasting (S6 Table) [152]. In a study by Gomez et al in Chile, the costs of outpatient treatment for those with severe wasting and/or bilateral pitting oedema or moderate wasting were $11 per child recovered per day for those 0 to 23 months old, and $5.89 per child recovered per day for those aged 24 to 71 months, from a societal perspective [175]. The household cost per child treated per day was $5.50 for children aged 0 to 71 months.

**Initiation of treatment in community settings.** In Chile, the costs of initiating treatment at a kindergarten for infants and children with severe wasting and/or bilateral pitting oedema or moderate wasting were $21 per child recovered per day for those 0 to 23 months old, and $15 per child recovered per day for those aged 24 to 71 months, from a societal perspective [175]. The household cost per child treated per day was $1.49 for children aged 0 to 71 months.

**Transfer from inpatient to outpatient.** In a study by Chapko et al in Niger, infants and children with severe wasting and/or bilateral pitting oedema or moderate wasting were first treated in the paediatric service of the national hospital and then randomised to nutritional rehabilitation in either a hospital inpatient setting or ambulatory setting [153]. Ambulatory treatment involved attendance at an ambulatory rehabilitation centre each day. From the provider perspective, the cost of transfer from inpatient to ambulatory treatment was $98, and that of transfer to another hospital inpatient setting was $217, per child treated (Table 5). This study, however, was conducted in 1994 and might not reflect the current treatment protocols.

## Study quality

Reporting quality was good for 45%, moderate for 45%, and low for 10% of studies (Table 3). There were nine areas that were most problematic. Only 14% of the included studies fully reported and justified the choice of discount rate(s) used for costs and outcomes. In addition, only 35% fully reported the dates of the estimated resource quantities and unit costs, methods for adjusting unit costs to the year of reported costs, or methods for converting costs into a common currency base and the exchange rate. The values, ranges, references, and probability distributions (where applicable) for all parameters and justification/ sources were fully reported by 39% of studies. Seventeen percent fully described all analytic methods supporting the evaluation. Thirty percent of studies fully characterised heterogeneity. Only 45% of

analyses using single study–based estimates of effectiveness fully described the design features of the single effectiveness study used and why this was a sufficient source of clinical effectiveness data. 48% of studies reported the mean values for the main categories of estimated costs and outcomes of interest for each intervention, as well as mean differences between the comparator groups. Only 9% of single study–based economic evaluation fully describes the effects of sampling uncertainty and methodological assumption on the reported estimates. For model-based analyses, 44% fully described and justified their model choice.

### Economic evidence profiles

We identified very few cost analyses (ranging from 1 to 5), and no cost-effectiveness analyses at all for the different models of care for the management of growth failure/faltering in infants <12 months of age; management of moderate wasting in infants and children <60 months of age; and management of moderate wasting and severe wasting and/or bilateral pitting oedema together in infants and children <60 months of age (Table 7). There were no direct comparisons between different models of care. These issues raise very serious concerns, hence the certainty of evidence for these three groups was judged as very low overall according to the GRADE ratings.

Evidence for the management of severe wasting and/or bilateral pitting oedema in infants and children <60 months of age suggests that costs for initiating treatment in community settings are the lowest, followed by initiating treatment in outpatient settings then transferring to community settings, then initiating treatment in outpatient settings, with initiating treatment in hospital settings being the most expensive. The certainty of the cost and cost-effectiveness evidence was, however, judged as very low for the following because only a few studies were identified, and there were no comparisons between different models of care: initiation of treatment in a combination of settings (3 cost and 0 cost-effectiveness studies); referral to treatment in an inpatient setting (4 cost and 1 cost-effectiveness studies); and transfer from outpatient treatment to treatment in a community setting (1 cost and 0 cost-effectiveness studies).

For initiation of treatment in community settings for severe wasting and/or bilateral pitting oedema in infants and children <60 months of age, 5 out of the 9 cost studies included comparisons with other settings. However, the cost-effectiveness results were based on one study and the uncertainty of the estimates were not explored. The certainty of the evidence was therefore judged as low. The certainty of the evidence for the initiation of treatment in outpatient settings for severe wasting and/or bilateral pitting oedema in infants and children <60 months of age was judged as moderate. There were 17 cost and 6 cost-effectiveness analyses. However, cost comparisons with other settings were based on only 5 studies, and for cost-effective analyses all comparisons were to a do nothing/ no programme scenario. This potentially limit the applicability of the evidence.

## Discussion

### Summary of principal findings

This review highlights glaring gaps in economic evidence to support decisions on models of care for growth failure/faltering in infants <12 months of age, and moderate wasting and severe wasting and/or bilateral pitting oedema in infants and children <60 months of age. The evidence remains inconclusive for most models of care. However, the evidence suggests that, for the treatment for severe wasting and/or bilateral pitting oedema in infants and children <60 months of age, the costs are lowest for initiating treatment in community settings, followed by initiating treatment in community and transferring to outpatient settings, initiating treatment in outpatients then transferring to community settings, initiating treatment in

**Table 7. Economic profiles.**

| Intervention | Resource allocation considerations | | Cost-effectiveness | | Quality (Based on CHEERS) | Applicability | Certainty | Other limitations/ comments |
|---|---|---|---|---|---|---|---|---|
| | *Number of studies* | *Costs* | *Number of studies* | *Summary of results* | | | | |
| **Management of growth failure/faltering in infants <12 months of age** | | | | | | | | |
| *Transfer from inpatient to community treatment* | 1 | Transfer from inpatient to community treatment in a medical placement home was more expensive per child treated than treatment in an inpatient setting [159]. | 0 | - | *Cost analysis:* Moderate | Directly applicable for <12 months age group Partially applicable for <6 months age group | No sensitivity analysis | No CEA studies comparing different settings |
| **Management of moderate wasting in infants and children <60 months of age** | | | | | | | | |
| *Initiation of treatment in community settings* | 5 | No direct comparisons made with other settings [163, 165–167, 169]. | 0 | - | *Cost analysis:* Moderate | Partially applicable: no direct comparisons with other settings | Sensitivity analysis only in one study but the study was evaluating increasing the amount of oil used in preparing corn-soy blend porridge in the same setting [169]. | No CEA studies comparing different settings |
| *Initiation of treatment in outpatient settings* | 2 | No direct comparisons made with other settings [158, 168]. | 0 | - | *Cost analysis:* Good | Partially applicable: no direct comparisons with other settings | Sensitivity analysis only in one study but the study evaluates different interventions in the same setting [158]. | No CEA studies comparing different settings |
| **Management of severe wasting and/or bilateral pitting oedema in infants and children <60 months of age** | | | | | | | | |
| *Initiation of treatment in community settings* | 9 | Costs per child treated/ recovered were lower overall when compared to outpatient [170–172] or hospital [160] treatment, or initiation in the community then referring to an inpatient setting [164]. | 1 | Cost-effective when compared to outpatient settings using WHO's GDP/ capita threshold [174] | *Cost analyses:* Moderate overall (1 low; 3 moderate; 5 good) *CEA:* Good overall | Directly applicable | No sensitivity analysis performed | Cost comparisons with other settings based on only 5 studies. Cost-effectiveness results are based on one study and the uncertainty of the estimated were not explored. |
| *Initiation of treatment in outpatient settings* | 17 | Costs per child treated/ recovered were lower overall compared to treatment in inpatient settings [148, 149]; but higher when compared to treatment in community settings [170–172] or initiating treatment in outpatient settings then transferring to community settings [149]. | 6 | Highly cost-effective when compared to a do nothing/ no programme scenario [147, 150, 154, 155, 164], using WHO's GDP/capita threshold Uncertainty on which strategy is most cost-effective when compared to initiation of treatment in community settings [170]. | *Cost analyses:* Moderate overall (1 low; 6 Moderate; 10 Good) *CEA:* Good overall | Directly applicable for cost analyses Partially applicable for CEA: comparisons were to a do nothing/ no programme scenario | Highly sensitive to projected number of deaths, costs of technical support, RUTF or to recover an additional child. | Cost comparisons with other settings based on only 5 studies. Five CEAs were accompanied by a PSA and/ or one-/two-way sensitivity analysis [147, 150, 155, 164, 170]. However, two studies seem to only include scenarios that would favour outpatient settings [147, 155]. |

**Table 7.** (Continued)

| Intervention | | Resource allocation considerations | Cost-effectiveness | | Quality (Based on CHEERS) | Applicability | Certainty | Other limitations/ comments |
|---|---|---|---|---|---|---|---|---|
| *Initiation of treatment in a combination of settings (community, outpatient and inpatient)* | 3 | Initiation of treatment in hospital and community settings costed more than initiation in community settings only [160]. | 0 | - | *Cost analyses: Moderate overall (1 Low; 1 Moderate and 1 Good)* | Directly applicable | No sensitivity analysis performed | Cost comparisons with other settings only from one study. No CEA studies comparing different settings |
| *Referral to treatment in an inpatient setting* | 4 | Referral from community and outpatient settings was of lower cost per child treated/ recovered compared to outpatient treatment [167, 168]. Referral from community settings was however more expensive that community treatment alone [164]. | 1 | Not cost-effective when compared to no treatment using WHO's GPD/capita threshold [164]. | *Cost-analyses: Good overall (2 good; 2 moderate) CEA: Good* | Directly applicable Partially applicable: for CEA as comparisons were to a no treatment scenario. | Highly sensitive to projected number of deaths. | CEA from one study. |
| *Transfer from outpatient treatment to treatment in a community setting* | 1 | Costs per child treated/ recovered were lower when compared to treatment in an outpatient or inpatient setting [149]. | 0 | - | *Cost analysis: Moderate* | Directly applicable | No sensitivity analysis | No CEA studies comparing different settings. |
| **Management of moderate wasting and severe wasting and/or bilateral pitting oedema together in infants and children <60 months of age** | | | | | | | | |
| *Initiation of treatment in outpatient settings* | 2 | One study made no direct comparisons made with other settings [152]. In one study, the daily societal costs per child recovered were lower for the 0–23 months age group, but higher for the 24 to 71 months age group, than for treatment initiation in community settings [175]. Daily household level costs per child treated were higher than treatment initiation in community settings [175]. | 0 | - | *Cost analysis: (1 good; 1 moderate)* | Directly applicable: direct comparison with community settings in one study. | Sensitivity analysis performed in one study for costs with greatest uncertainty but not clear which ones and their impact [152]. | No CEA studies comparing different settings |

*(Continued)*

**Table 7.** (Continued)

| Intervention | Resource allocation considerations | Cost-effectiveness | Quality (Based on CHEERS) | Applicability | Certainty | Other limitations/comments |
|---|---|---|---|---|---|---|
| *Initiation of treatment in community settings* | 1 — In one study, the daily societal costs per child recovered were higher for the 0–23 months age group, but lower for the 24 to 71 months age group, than for treatment initiation in outpatient settings [175]. Daily household level costs per child treated were lower than treatment initiation in community settings [175]. | 0 — | *Cost analysis:* Moderate | Directly applicable | No sensitivity analysis | No CEA studies comparing different settings. |
| *Transfer from inpatient to outpatient* | 1 — the cost of transfer from inpatient to outpatient treatment was lower than that of transfer to another hospital inpatient setting, per child treated [153]. | 0 — | *Cost analysis:* Moderate | Directly applicable | Performed for loss to follow-up; per protocol and anthropometric outcomes. These analyses did not substantially differ from the main results. | No CEA studies comparing different settings |

outpatient settings, and lastly initiating treatment in inpatient settings. Our findings also suggest that, for infants and children <60 months of age with severe wasting and/or bilateral pitting oedema, initiation of treatment in outpatient settings is highly cost-effective when compared to a do-nothing/ no programme scenario. This is in line with another review that also concluded that outpatient facility-based care for child wasting was highly cost-effective [10]. However, the strength of this evidence is limited due to a lack of comparisons with other alternatives.

Costs seem to vary widely within and across regions. For example, the initiation of treatment for severe wasting and/or bilateral pitting oedema in a community setting for those <60 months of age costs between $202 and $757 in Africa, and $206 to $7,987 in South-East Asia, per child treated from a provider perspective. Even within the same country, costs seem to vary widely. For example, the cost of initiating treatment for severe wasting and/or bilateral pitting oedema in outpatient settings in Mali varied between $103 and $1208 per child treated from a provider perspective. This suggests a strong influence of contextual factors that might limit the transferability of costs data from one setting to another [177]. Cost drivers that have been highlighted in this respect include price levels, population density, the scale of the programme, the underlying health of the population, existing health infrastructure, and implementer capacity [9, 10, 150, 173, 177]. For example, a programme situated in a population where the number of malnourished children is high is likely to experience lower costs per child treated or recovered than one in a population where these numbers are low [10]. This is because the costs, for example, indirect costs, will be divided among more children. A few of the studies included in our review performed sensitivity analysis and they reported that their results were highly sensitive to projected number of deaths, costs of technical support, RUTF or to recover an additional child [147, 150, 155, 164, 170].

Evidence on this topic is dominated by cost or cost-efficiency analyses, rather than cost-effectiveness analyses that can inform decisions on how to maximise outcomes and minimise opportunity costs. Many of the included studies also took a provider perspective despite undernutrition being a multisectoral problem resulting in resource consumption and impacting on outcomes in other sectors such as the education sector [178]. Community and household costs have been largely ignored: for example, only seven studies included productivity losses [161, 162, 165, 169–172], and four included transport costs incurred by families/caregivers [161, 165, 171, 172]. This is despite the fact that these costs can be high and potentially catastrophic, particularly for poor households, those without a reliable source of income, or where most of the costs are paid for out-of-pocket. In addition, these costs are very well-known barriers to accessing treatment and can differ substantially between different treatment models. For example, studies have suggested that community-based treatment for severe wasting and/or bilateral pitting oedema in children under 5 years of age can decrease household costs by six times when compared to inpatient treatment [164], and by three times compared to outpatient facility-based care [170, 171]. A wide range of outcomes were used, with inconsistencies in how they were defined, making comparisons across studies difficult.

## Strengths and limitations of the review

To our knowledge, this is the first comprehensive systematic review that focuses on the cost and cost-effectiveness of different models of care for the management of infants <12 months of age with growth faltering/failure, and infants and children aged <60 months of with moderate wasting and severe wasting and/or bilateral pitting oedema. We used a very comprehensive search strategy to identify both peer-reviewed journal articles and grey literature. We included all eligible study reports regardless of language, date of publication or the country in which the

study was conducted. We also clearly defined our key variables of interest, i.e., population, setting and type of care, to ensure consistency and transparency in the way the studies were classified. All costs were converted to 2020 US dollar costs to facilitate comparisons across countries. Nevertheless, heterogeneity in perspectives of the evaluations, included costs, costing methods, outcomes and how these were defined and measured meant that we were not able to pool results or make comparisons across studies.

## Recommendations for policy and practice

Costs vary widely according to context and, therefore, the costs for the different models of care are also likely to vary greatly across countries. For most of the models of care explored in this systematic review, there is not enough cost-effectiveness evidence to inform recommendations for policy and practice. For severe wasting and/or bilateral pitting oedema in infants and children aged <60 months, initiation of treatment in outpatient settings can be recommended. There is no reason to believe that this will be different for the age group 6 to 59 months, which is the focus of the WHO recommendations for child wasting. However, most of the studies compared initiation of treatment in outpatient setting to do nothing/ no programme scenarios rather than to other settings or models of care. This may limit the applicability of these findings in real-world settings.

## Recommendations for future research

There is a need for further research on both the costs and cost-effectiveness of different treatment models for the management of growth failure/faltering in infants <12 months of age, and the management of moderate wasting and severe wasting and/or bilateral pitting oedema in infants and children aged <60 months. The research should compare the different alternatives. There is also a need for the standardisation of outcomes, including definitions and assessment methods. Some researchers have suggested that, as a minimum, studies should include the cost per child recovered [10]. This can be calculated using routine programme data. In addition, more comprehensive outcomes such as the DALY would allow comparisons with child health interventions: this is important for decision-makers. To be able to identify the main costs drivers, including contextual determinants, there is a need to develop a minimum set of costs that have to be included when conducting cost or cost-effectiveness analyses in this research area. Capturing the societal costs, rather than just the provider costs is also important due to the multi-sectoral nature of undernutrition [9, 10]. It will allow for decisions that account for the costs and cost savings to other sectors as well as the beneficiary's households.

## Conclusions

There is very limited economic evidence to inform policy and practice on the setting of treatment initiation, referral, transfer and discharge of: 1) infants <12 months of age with growth faltering/failure; and 2) infants and children aged <60 months with moderate wasting. For infants and children aged <60 months with severe wasting and/or bilateral pitting oedema, evidence suggests that initiation of treatment in outpatient settings is highly cost-effective. However, the applicability of these findings in real-world settings could be limited as most of the comparisons are to do nothing/ no programme scenarios rather than to other settings or models of care.

## Supporting information

**S1 File. PRISMA checklist.**
(PDF)

**S2 File. Key definitions.**
(PDF)

**S3 File. Search strategies.**
(PDF)

**S4 File. Data extraction categories and variables.**
(PDF)

**S5 File. Characteristics of excluded studies.**
(PDF)

**S6 File. Study definitions for moderate wasting, severe wasting and/or bilateral pitting oedema, or growth failure/faltering and inclusion/exclusion criteria.**
(PDF)

**S1 Table. Cost analysis results for management of growth failure/faltering in infants <12 months of age.**
(DOCX)

**S2 Table. Cost analysis results for management of moderate wasting in infants and children <60 months of age.**
(DOCX)

**S3 Table. Cost-effectiveness analysis results for the management of moderate wasting in infants and children <60 months of age.**
(DOCX)

**S4 Table. Cost analysis results for the management of severe wasting and/or bilateral pitting oedema in infants and children <60 months of age.**
(DOCX)

**S5 Table. Cost-effectiveness analysis results for the management of severe wasting and/or bilateral pitting oedema in infants and children <60 months of age.**
(DOCX)

**S6 Table. Cost analysis results for the management of moderate wasting and severe wasting and/or bilateral pitting oedema together in infants and children <60 months of age.**
(DOCX)

## Acknowledgments

We would like to thank Allison Daniel, Jaden Bendabenda, Kirrily de Polnay, Zita Weise Prinzo, Celeste Naude, Michael McCaul, and the World Health Organization Guideline Development Group for their input on the methodology and interpretation of results.

## Author Contributions

**Conceptualization:** Noreen Dadirai Mdege, Sithabiso D. Masuku, Nozipho Musakwa, Mphatso Chisala, Farhad Shokraneh.

**Data curation:** Noreen Dadirai Mdege, Sithabiso D. Masuku, Nozipho Musakwa, Mphatso Chisala, Farhad Shokraneh.

**Formal analysis:** Noreen Dadirai Mdege, Sithabiso D. Masuku, Nozipho Musakwa, Mphatso Chisala, Ernest Ngeh Tingum, Micheal Kofi Boachie.

**Funding acquisition:** Noreen Dadirai Mdege, Sithabiso D. Masuku, Nozipho Musakwa, Mphatso Chisala, Farhad Shokraneh.

**Investigation:** Noreen Dadirai Mdege, Sithabiso D. Masuku, Nozipho Musakwa, Mphatso Chisala.

**Methodology:** Noreen Dadirai Mdege, Sithabiso D. Masuku, Nozipho Musakwa, Mphatso Chisala, Farhad Shokraneh.

**Project administration:** Noreen Dadirai Mdege.

**Resources:** Noreen Dadirai Mdege.

**Supervision:** Noreen Dadirai Mdege.

**Validation:** Noreen Dadirai Mdege, Sithabiso D. Masuku, Nozipho Musakwa, Mphatso Chisala, Ernest Ngeh Tingum, Micheal Kofi Boachie.

**Visualization:** Noreen Dadirai Mdege, Sithabiso D. Masuku, Nozipho Musakwa, Mphatso Chisala.

**Writing – original draft:** Noreen Dadirai Mdege.

**Writing – review & editing:** Noreen Dadirai Mdege, Sithabiso D. Masuku, Nozipho Musakwa, Mphatso Chisala, Ernest Ngeh Tingum, Micheal Kofi Boachie, Farhad Shokraneh.

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
