## [Decision Letter · Decision Letter 0]

30 Aug 2023

PGPH-D-23-01064

Costs and cost-effectiveness of treatment setting for children with wasting, oedema and growth failure/faltering: a systematic review

Dear Dr. Mdege,

Thank you for submitting your manuscript to PLOS Global Public Health. After careful consideration, we feel that it has merit but does not fully meet PLOS Global Public Health’s publication criteria as it currently stands. Therefore, we invite you to submit a revised version of the manuscript that addresses the points raised during the review process.

We look forward to receiving your revised manuscript.

Kind regards,

Sirshendu Chaudhuri, MD, DPH

Academic Editor

Journal Requirements:

Additional Editor Comments (if provided):

Reviewers' comments:

Reviewer's Responses to Questions

**Comments to the Author**

1. Does this manuscript meet PLOS Global Public Health’s publication criteria? Is the manuscript technically sound, and do the data support the conclusions? The manuscript must describe methodologically and ethically rigorous research with conclusions that are appropriately drawn based on the data presented.

Reviewer #1: Yes

Reviewer #2: Yes

2. Has the statistical analysis been performed appropriately and rigorously?

Reviewer #1: Yes

Reviewer #2: Yes

3. Have the authors made all data underlying the findings in their manuscript fully available (please refer to the Data Availability Statement at the start of the manuscript PDF file)?

Reviewer #1: Yes

Reviewer #2: Yes

4. Is the manuscript presented in an intelligible fashion and written in standard English?

Reviewer #1: Yes

Reviewer #2: Yes

5. Review Comments to the Author

Reviewer #1: I appreciate the authors for the tremendous work in completing the review. I am glad that the care giver perspective and the societal perspective are taken into consideration. The review fulfilled the reporting guidelines appropriately.

Reviewer #2: Overall, the article is good in reporting the effectiveness of treatment settings for children with wasting, oedema, and growth failure/faltering in terms of both cost and outcomes.

There are very few areas which require minor revision:

1.In table 1, the exclusion criteria mentioned are just the opposite to the statements mentioned in inclusion criteria which can’t be so and need to be re-framed

2.Line number 215, 396, 552 to be checked for typo

3.Figure 1 reference to 115 records exclusion in search results

6. PLOS authors have the option to publish the peer review history of their article (what does this mean?). If published, this will include your full peer review and any attached files.

**Do you want your identity to be public for this peer review?** For information about this choice, including consent withdrawal, please see our Privacy Policy.

Reviewer #1: **Yes: **DR. C. SRAVANA DEEPTHI

Reviewer #2: No

---

## [Editor Report · Decision Letter 1]

10 Oct 2023

Costs and cost-effectiveness of treatment setting for children with wasting, oedema and growth failure/faltering: a systematic review

PGPH-D-23-01064R1

Dear Dr. Mdege,

We are pleased to inform you that your manuscript 'Costs and cost-effectiveness of treatment setting for children with wasting, oedema and growth failure/faltering: a systematic review' has been provisionally accepted for publication in PLOS Global Public Health.

Best regards,

Sirshendu Chaudhuri, MD, DPH

Academic Editor